# A Descriptive and Normative Theory of Human Beliefs in RLHF

**Sylee Dandekar**                                                  *sdandekar@umass.edu*
*College of Information and Computer Sciences*
*University of Massachusetts Amherst*

**Shripad V. Deshmukh**                                         *svdeshmukh@umass.edu*
*College of Information and Computer Sciences*
*University of Massachusetts Amherst*

**Frank Chiu**                                                          *fchiu@umass.edu*
*College of Information and Computer Sciences*
*University of Massachusetts Amherst*

**W. Bradley Knox**                                          *bradknox@cs.utexas.edu*
*Department of Computer Science*
*The University of Texas at Austin*

**Scott Niekum**                                                    *sniekum@umass.edu*
*College of Information and Computer Sciences*
*University of Massachusetts Amherst*

**Reviewed on OpenReview:** *https://openreview.net/forum?id=YdWOKZwPeT*

## Abstract

Human preferences in RLHF are typically modeled as a function of the human's reward function or corresponding optimal state-action values. In this work, we propose that human beliefs about the capabilities of the agent being trained also play a key role in preference generation. We examine two questions related to this hypothesis, one descriptive and one normative, respectively: Do human labelers' beliefs about agent capabilities affect the preferences that they provide? And what is the ideal set of beliefs about an agent— and resulting preferences—for humans to have? We propose a new preference model that incorporates human beliefs and provide a normative theory that bounds the error on the final learned policy based on the *mismatch* between the human's beliefs and an idealized set of beliefs. We then confirm via a human study that beliefs about agent capabilities do, in fact, significantly affect preferences and can be influenced through simple interventions. Additionally, we empirically show through synthetic experiments that it is often suboptimal for human preference labelers to assume agent optimality. Collectively, these results theoretically and empirically demonstrate how reducing the mismatch between human beliefs and agent capabilities can lead to more performant RLHF and point toward new best practices for RLHF practitioners.

## 1    Introduction

Reinforcement learning from human feedback (RLHF) is one of the main tools used to align powerful AI systems (Kaufmann et al., 2023). Alignment is important for numerous reasons, such as minimizing unintentional harm, ensuring safety and control, and increasing public trust and legal compliance (Ji et al., 2023; Amodei et al., 2016). In order to use RLHF for alignment more effectively, researchers require high-quality data from humans that express rational preferences. This begs the question: *How should rationality of*

*preferences be defined within RLHF?* Standard approaches to RLHF assume that humans provide preferences based only on what actually happened in the trajectories, i.e., the return-based interpretation (Christiano et al., 2017; Ziegler et al., 2019; Ouyang et al., 2022; Brown et al., 2019). This partial return-based approach does not consider events that *could have* occurred while taking a risky action or what might happen after the end of the trajectory segment. Knox et al. (2024) show that the regret-based model addresses this by looking at the quality of actions taken, rather than just the summed reward of the observed outcomes.

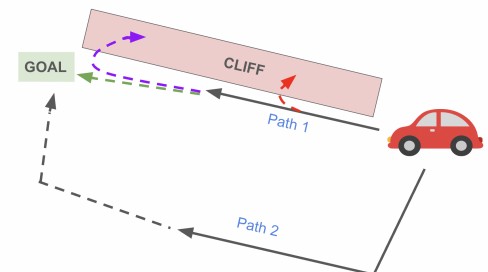

Figure 1: Illustration of scenario in which preference for a more optimal partial path can lead to a worse post-RLHF policy. *(Path 1)*: The car drives along the edge of a cliff but straight to the destination. This path takes less time but requires greater capabilities (both during and after) to avoid catastrophic outcomes. *(Path 2)*: The car takes a longer path far away from the cliff, reducing risk. The dotted lines indicate different possible paths that could occur when executing a post-RLHF policy: red indicates the possibility that the agent may drive off the cliff if it fails to perfectly imitate Path 1; purple indicates a suboptimal policy starting from the final state of Path 1 that also drives off the cliff; green indicates a policy that is able to safely reach the goal. If the post-RLHF policy would induce the red or purple trajectories under a Path 1 preference, then it would be better for the labeler to have preferred Path 2, despite the longer travel time.

However, even the regret-based approach may not fully capture the factors that humans may consider when giving preferences. The regret model assumes that humans judge the "goodness" of an action as a function of the optimal advantage function $A^*$ that the agent should ideally be able to achieve under the assumed MDP, given enough data. But what if humans do not assume optimality when judging action quality, but instead rely on their beliefs about the agent's capabilities—in other words $A^{\pi_{\text{belief}}}$ for some imagined policy $\pi_{\text{belief}}$ rather than $A^*$?

For example, consider the scenario in Figure 1, in which an agent must choose between a shorter cliff-side path to a goal or a longer, safer path far away from the cliff. Path 1 is preferable if the agent can reproduce the trajectory noiselessly and also behave optimally afterwards. However, if the agent is suboptimal, then it risks falling off the cliff when taking this path or afterwards. Conversely, Path 2 is better for a suboptimal agent, but overly conservative for an optimal agent. Suppose that a human labeler provides a preference for Path 1 based on the incorrect belief that the agent will behave optimally post-RLHF. If the agent is trained on these preferences and is suboptimal post-RLHF, this can lead to catastrophic outcomes.

This motivates the questions: *Should a rational preference labeler consider agent capabilities? And what is a normative ideal for labeler beliefs about agent capabilities?* There are a variety of factors that can affect an agent's capabilities. Factors such as limited training data, incomplete or noisy data, or policy parameterization could limit the space of learnable policies. This limitation of learnable policies, by extension, limits agent performance. Furthermore, if the agent is a robot, physical limitations may also limit agent performance beyond what a human might assume. Given a fixed set of trajectory pairs, we show that the normative ideal in regret-based RLHF is for labelers to provide preferences with respect to a quantity that is related to the best possible post-RLHF policy under any labeling—as opposed to the standard assumption of absolute optimality.

Our contributions are as follows. (1) We illustrate the importance of beliefs about agent capabilities by showing an example where poor agent-labeler agreement leads to suboptimal post-RLHF policies. (2) We define a normative ideal of beliefs and theoretically investigate the impact of deviations from this ideal on the expected return of the post-RLHF policy. (3) We empirically examine the effect of the magnitude of

deviation from the ideal belief by manipulating it directly in synthetic experiments, showing that the highest performance typically results from beliefs closest to our theoretical normative ideal. (4) Finally, we confirm in a human study that human beliefs about the agent's capabilities affect preferences. Specifically we show that it is possible to change human preferences in a statistically significant manner by changing their beliefs through priming. This study also points to potential paths forward for aligning labeler beliefs with agent capabilities.

## 2 Preliminaries

A Markov Decision Process (MDP) (Puterman, 2014; Sutton & Barto, 1998) is specified by a tuple $(\mathcal{S}, \mathcal{A}, P, \gamma, \mu, r)$. Here, $\mathcal{S}$ is the set of possible states, and $\mathcal{A}$ is the set of all possible actions. $P$ is the transition function $P : \mathcal{S} \times \mathcal{A} \to \Delta(\mathcal{S})$, a probability distribution over the next state given the current state and action. $\gamma$ is the discount factor, $\mu$ is the start state distribution, and $r : \mathcal{S} \times \mathcal{A} \times \mathcal{S} \to \mathbb{R}$ is the scalar reward function. $r_t$ refers to the reward received at time $t$ for taking action $a_t$ at state $s_t$ and reaching $s_{t+1}$. Terminal states have a reward of 0, unless otherwise specified, upon reaching them and a reward of 0 forever after. Throughout this paper, $r$ refers to the ground-truth reward function, from which preferences are generated.

A policy $\pi : \mathcal{S} \to \Delta(\mathcal{A})$, specifies the probability of taking an action in a given state. $V_r^\pi$ is the state-value function for $\pi$ under the reward function $r$, defined as: $V_r^\pi(s) = E_\pi[\sum_{t=0}^\infty \gamma^t r(s_t, a_t, s_{t+1})|s_0 = s]$. $Q_r^\pi$ is the state-action value function: $Q_r^\pi(s, a) = E_\pi[r(s, a, s') + \gamma V_r^\pi(s')]$. The advantage function with respect to $\pi$ is defined as $A_r^\pi(s, a) = Q_r^\pi(s, a) - V_r^\pi$. $J_r^\pi = E_\pi[\sum_{t=0}^\infty \gamma^t r(s_t, a_t, s_{t+1})|s_0 \sim \mu]$ denotes the policy's expected discounted return. We use $A^\pi$, $V^\pi$, $Q^\pi$, and $J^\pi$ to denote $A_r^\pi$, $V_r^\pi$, $Q_r^\pi$, and $J_r^\pi$ respectively. An optimal policy $\pi^*$ is a policy for which $V_r^{\pi^*}(s) \geq V_r^\pi(s)$ for all states and all policies. We use $A^*$, $V^*$, and $Q^*$ to denote $A_r^{\pi^*}$, $V_r^{\pi^*}$, and $Q_r^{\pi^*}$ respectively.

**Trajectories:** $\sigma$ refers to a trajectory with starting state $s_0^\sigma$. The length of a trajectory $|\sigma|$ refers to the number of transitions in the trajectory. Each trajectory has $|\sigma + 1|$ states and $|\sigma|$ actions. In the context of our setting, trajectories do not contain any reward information as rewards are unknown and are to be inferred from preferences.

**Preference Dataset:** We have a preference dataset $\mathcal{D}_{\text{pref}} = \{(\sigma_i^+, \sigma_i^-) \mid i = 1, 2, ..., n\}$ where each pair of trajectories $(\sigma_i^+, \sigma_i^-)$ are of equal length and $\sigma_i^+ \succ \sigma_i^-$ denotes that the first trajectory is strictly preferred over the second.

**Preference Models:** The pairwise preferences in $\mathcal{D}_{\text{pref}}$ are assumed to be sampled from an underlying distribution $P(\sigma^+ \succ \sigma^-)$ which is modeled using preference model (e.g. the partial-return model (Christiano et al., 2017) or the regret model (Knox et al., 2024)) obeying ground truth reward $r$. The partial-return preference model is given by:

$$P(\sigma^+ \succ \sigma^-) = \frac{\exp(\alpha \sum_{\sigma^+} \gamma^t r_t)}{\exp(\alpha \sum_{\sigma^+} \gamma^t r_t) + \exp(\alpha \sum_{\sigma^-} \gamma^t r_t)} \tag{1}$$

and the regret-based preference model is given by :

$$P^*(\sigma^+ \succ \sigma^-) = \frac{\exp(\alpha \sum_{\sigma^+} \gamma^t A^*(s_t, a_t))}{\exp(\alpha \sum_{\sigma^+} \gamma^t A^*(s_t, a_t)) + \exp(\alpha \sum_{\sigma_-} \gamma^t A^*(s_t, a_t))} \tag{2}$$

where $\alpha$ is an inverse temperature coefficient.

## 3 Related Work

**Reinforcement Learning from Human Feedback.** RLHF methods traditionally follow a two-step learning framework: (1) learning a reward function using preference data and (2) using RL to perform policy optimization (Christiano et al., 2017; Ouyang et al., 2022; Brown et al., 2019). Direct methods remove the RL step from this traditional approach by directly learning policies from preferences but make the same

preference modeling assumptions as standard return-based RLHF (Munos et al., 2024; Rafailov et al., 2024; Hejna et al., 2024).

**Modeling human beliefs in learning.** Prior work has aimed to model human beliefs in learning settings beyond RLHF. Reddy et al. (2018) state that the reason demonstrators deviate from near-optimal actions is because of a misunderstanding of the environment dynamics and propose an algorithm to estimate these misunderstandings. Gong & Zhang (2020) assume that humans do not maintain a correct belief about dynamics when giving preferences and propose a method to infer both the reward function and the human's beliefs about dynamics. Marklund & Roy (2024) aim to separate goals from potentially incorrect human beliefs about environmental transition dynamics. Chan et al. (2021) demonstrate that human irrationality, when correctly modeled, can be an asset to reward inference.

**Biases in human data collection.** While RLHF assumes that humans offering preferences do so rationally and in alignment with their long-term objectives, biases in human survey data have been studied across various fields of research (Kaufmann et al., 2023; Kahneman, 2003; Köszegi & Rabin, 2008). It is widely recognized that human decisions can be swayed by biases due to irrationality, incomplete, or inaccurate information (Gilovich T, 2002). Köszegi & Rabin (2008) state that mistakes are systematic and can be revealed by behavior and show that previously seen small amounts of non representative data can bias preferences. Previous psychology-aware RLHF literature investigate the impact of biasing effects on preferences. Lichtenstein & Slovic (2006) discuss how preferences can be formed at the same time as the process of preference elicitation. Personality, emotions, social connections, and decision biases (such as decoy effects, serial position effects, anchoring effects, framing effects, and group think) can also affect preferences (Tran et al., 2021; Atas et al., 2021). Ziegler et al. (2019) discuss the importance of alignment between researcher goals and the labeler's labels on open ended summarization and sentiment analysis tasks.

## 4   Theory: Effects of agent-labeler disagreement

In this section, we aim to bound the impact of agent-labeler disagreement on expected returns of an RLHF-trained policy. We first formally define the notions of agent capability belief and agent-labeler disagreement. We then present an example MDP where agent-labeler disagreement leads to agents learning suboptimal policies. Finally, we provide a formal bound on regret of the post-RLHF policy loss as a function of agent-labeler disagreement.

Under the regret-based model of preferences, the labeler is assumed to provide preferences based on the advantage of actions under the optimal advantage function $A^*$. However, a real-world agent's performance after RLHF will not typically be optimal and instead will be limited by the amount of preference data, training time, or the learning algorithm itself. We will show that if humans do, in fact, generate preferences via the regret preference model, it can lead to catastrophically bad post-RLHF policies. This is due to the assumption of optimality following the trajectory segments for which preferences are given. For example, in Figure 1, the actual advantage of choosing Path 1 depends on whether or not the car will behave optimally in future time steps during execution. Thus, when providing preferences, we posit that the quality of an action ought to be related to the agent's post-RLHF advantage function, rather than assuming optimality.

Formally, we denote the labeler's estimate of the agent's post-RLHF policy as $\pi_{\text{belief}}$. $Q^{\pi_{\text{belief}}}$ and $A^{\pi_{\text{belief}}}$ denote the corresponding state-action value function and advantage function respectively. We propose a new preference model, similar to the regret-based preference model, but which aims to explicitly capture how humans might incorporate such beliefs. We call this new preference model the **belief-based preference model**:

$$P^{\text{belief}}(\sigma^+ \succ \sigma^-) = \frac{\exp(\alpha \sum_{\sigma^+} \gamma^t A^{\pi_{\text{belief}}}(s_t, a_t))}{\exp(\alpha \sum_{\sigma^+} \gamma^t A^{\pi_{\text{belief}}}(s_t, a_t)) + \exp(\alpha \sum_{\sigma^-} \gamma^t A^{\pi_{\text{belief}}}(s_t, a_t))} \tag{3}$$

where $\alpha$ is an inverse temperature coefficient. The remainder of the paper will examine the theoretical and empirical consequences of humans providing belief-based preferences under this model, as well as establish evidence via a human study that human preferences are, in fact, influenced by beliefs about agent capabilities. In order to do so, we must first formally define agent capabilities, as well as agent-labeler disagreement.

We assume that the human has a belief about the capabilities of the agent, which they use to generate preferences. We define this belief over capabilities in terms of a Q-function that represents the quality of an imagined agent's policy:

**Definition 4.1.** ***Agent capability belief*** is defined as the state-action value function, $Q^{\pi_{\text{belief}}}$, used to generate preferences in the belief-based preference model proposed in equation 3, via the associated $A^{\pi_{\text{belief}}}$.

In other words, preferences are generated based on the assumption that, after the completion of the preferred partial trajectory, the agent will follow $\pi_{\text{belief}}$, whose state-action value function is $Q^{\pi_{\text{belief}}}$.

One might then ask: what $\pi_{\text{belief}}$ might a human imagine when providing preferences? The standard regret-based model assumes that the human imagines optimal behavior, but our human study in Section 6 demonstrates that preferences are subject to priming effects, indicating that the human's beliefs about agent capabilities may be more closely related to the perceived current policy of the agent, or a prior over similar agents formed from past experiences, or perhaps an estimated performance ceiling for such agents.

Thus, another key question is: What is the ideal $Q^{\pi_{\text{belief}}}$ for a human to hold? Assume that the human is asked to provide preferences over a fixed, finite set of pairs of partial trajectories. The normative ideal is for the human to provide preferences in a manner that results in the highest expected return of the post-RLHF policy. We denote any policy that achieves this maximum as $\pi_{\text{post}}^*$. Note that this policy is not necessarily optimal for the MDP; it is simply the best that can be achieved after an RLHF step under any assignment of preferences over the fixed set of trajectory pairs. Further, there exist a set of beliefs that lead to such maximizing policies when preferences are generated according to them. We denote these as $Q^{\pi_{\text{belief}}^*}$. This optimal belief may require oracle knowledge to generate (e.g. environmental dynamics or learning rule), but it is the normative ideal.

We stress that $Q^{\pi_{\text{belief}}^*}$ is a theoretical reference point: we use it to define agent-labeler disagreement and to derive the bound in Section 4.2, and do not propose computing or estimating it in practice. The direction of interpretation also matters: $Q^{\pi_{\text{belief}}^*}$ is not derived from $\pi_{\text{post}}^*$; rather, it represents accurate beliefs about the agent's capabilities, and it is labelers holding such beliefs that yields preferences recovering $\pi_{\text{post}}^*$.

Finally, note that if an infinite number of (noiseless) preferences are to be given and any policy is representable by the agent, this implies that the labeler ought to have a belief of optimal behavior, normatively; in this case, $\pi_{\text{post}}^*$ is only suboptimal when a finite number of preferences are given. However, if preferences are noisy or the agent cannot represent an optimal policy (e.g. a weak function approximator), then this may not be true.

We now define agent-labeler disagreement, which captures the more realistic scenario of a human labeler providing preferences under some agent capability belief $Q^{\pi_{\text{belief}}}$ that is different from normative ideal, $Q^{\pi_{\text{belief}}^*}$.

**Definition 4.2.** We define the ***agent-labeler disagreement*** at a state-action pair as the magnitude of difference between an optimal belief $Q^{\pi_{\text{belief}}^*}$ and a human labeler's belief $Q^{\pi_{\text{belief}}}$ given by $|Q^{\pi_{\text{belief}}^*}(s,a) - Q^{\pi_{\text{belief}}}(s,a)|$. When there exist multiple $Q^{\pi_{\text{belief}}^*}$, the minimum disagreement is used.

## 4.1 A case study

Here we present an example to illustrate the effect of agent-labeler disagreement. Consider the simple MDP shown in Figure 2, which is adapted from an example in Knox et al. (2024) to capture the risk of poor decision-making *after* an action.[1] $s_0$ is the start state, and arrows show deterministic transitions towards the three terminal states on the far right. All rewards are 0 unless otherwise noted.

Suppose a labeler is asked to provide preferences between two partial trajectories: $(s_0, a_{\text{risk}}, s_{\text{risk}})$ and $(s_0, a_{\text{safe}}, s_{\text{safe}})$, and their beliefs are such that $A^{\pi_{\text{belief}}}(s_0, a_{\text{risk}}) > A^{\pi_{\text{belief}}}(s_0, a_{\text{safe}})$. Then, under the belief-based preference model, they would tend to label $(s_0, a_{\text{risk}}, s_{\text{risk}}) \succ (s_0, a_{\text{safe}}, s_{\text{safe}})$. If the agent-labeler disagreement is such that $A^{\pi_{\text{belief}}^*}(s_0, a_{\text{risk}}) \leq A^{\pi_{\text{belief}}^*}(s_0, a_{\text{safe}})$—which would occur if $\pi_{\text{belief}}^*(s_{\text{risk}}, a_{\text{lose}}) > 0.5$—then obeying the preference above results in losing more often than winning during execution, resulting in a learned policy that has lower expected return than taking $a_{\text{safe}}$.

---

[1]Note that decision-making itself may not appear to be a risk from the agent's perspective, since the agent controls its own actions at subsequent states. However, even deterministically chosen actions can be risky from the labeler's perspective because the labeler is only able to influence the agent at the current state.

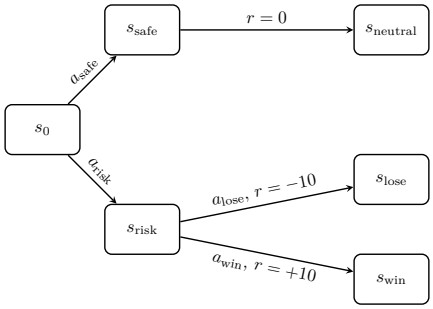

Figure 2: An MDP in which a safe and a risky action are available. If $A^{\pi_{\text{belief}}}(s_0, a_{\text{risk}}) > A^{\pi_{\text{belief}}}(s_0, a_{\text{safe}})$ when $A^{\pi^*_{\text{belief}}}(s_0, a_{\text{risk}}) \leq A^{\pi^*_{\text{belief}}}(s_0, a_{\text{safe}})$, labelers will erroneously prefer the partial trajectory $(s_0, a_{\text{risk}}, s_{\text{risk}})$ over $(s_0, a_{\text{safe}}, s_{\text{safe}})$ under the belief-based preference model in Equation 3.

Conversely, suppose the labeler's beliefs are such that $A^{\pi_{\text{belief}}}(s_0, a_{\text{risk}}) < A^{\pi_{\text{belief}}}(s_0, a_{\text{safe}})$. Then under the belief-based preference model, they would prefer $(s_0, a_{\text{safe}}, s_{\text{safe}})$ over $(s_0, a_{\text{risk}}, s_{\text{risk}})$. Obeying this preference results in a return of 0 instead of a higher return during execution if $A^{\pi^*_{\text{belief}}}(s_0, a_{\text{risk}}) > A^{\pi^*_{\text{belief}}}(s_0, a_{\text{safe}})$. Thus, we can see that agent-labeler disagreements can lead to suboptimal preferences and resulting policies.

## 4.2 Formal bounds on impact of agent-labeler disagreements

We now bound the impact on the expected return of the RLHF-trained policy under agent-labeler disagreements. We work in the tabular, pairwise-comparison setting standard in theoretical analyses of RLHF (Zhu et al., 2023; Wang et al., 2023), which admits a clean characterization of how belief disagreement translates into performance loss. We first examine a setting where the labeler provides a preference under an agent-labeler disagreement at a single state-action pair, and later discuss how this analysis extends to multiple disagreements.

**Theorem 4.3.** *Consider an RLHF setup where a human labeler is provided with a finite number of pairs of single-transition trajectories to label. Let $Q^{\pi^*_{belief}}$ be an optimal agent capability belief (i.e. the normative ideal belief) for generating preferences using belief-based model (Eqn 3) over these pairs of trajectories, and let $\pi^*_{post}$ be the corresponding post-RLHF policy. Let $Q^{\pi_{belief}}$ be the human labeler's belief with an agent-labeler disagreement of $\delta$ at a state-action pair $(s', a')$ w.r.t. $Q^{\pi^*_{belief}}$. Let the post-RLHF policy trained with preferences given under $Q^{\pi_{belief}}$ be $\pi^{\delta}_{post}$. Assume that preferences are noiseless under the belief-based model (i.e. $\alpha \to \infty$). Further assume that the policy is tabular and RLHF produces a deterministic policy that respects all the preferences. We then have:*

$$J_{\pi^{\delta}_{post}} \geq J_{\pi^*_{post}} - \frac{\delta}{1 - \gamma}, \tag{4}$$

*where $J_{\pi^*_{post}}$ and $J_{\pi^{\delta}_{post}}$ are the expected returns of $\pi^*_{post}$ and $\pi^{\delta}_{post}$ respectively.*

*Proof.* Consider preferences given over single-transition trajectories via the belief-based model (Eqn. 3). When RLHF respects the given preferences, resulting in a deterministic post-RLHF policy $\pi_{\text{post}}$, we have $\pi_{\text{post}}(s) = \arg\max_a Q^{\pi_{\text{belief}}}(s, a)$. Under perfect agent-labeler agreement, we are given the post-RLHF policy as $\pi^*_{\text{post}}$.

Now, we examine the impact of agent-labeler disagreement at $(s', a')$ on the post-RLHF policy. We have disagreement of

$$\delta = \left| Q^{\pi^*_{\text{belief}}}(s', a') - Q^{\pi_{\text{belief}}}(s', a') \right|. \tag{5}$$

When this disagreement grows such that $\delta > Q^{\pi_{\text{belief}}}(s', \pi_{\text{post}}^*(s')) - Q^{\pi_{\text{belief}}}(s', a')$, action $a'$ becomes the most preferred action. This results in a new post-RLHF policy $\pi_{\text{post}}^\delta$ as follows:

$$\pi_{\text{post}}^\delta(s) = \begin{cases} \pi_{\text{post}}^*(s) & \text{if } s \neq s' \\ a' & \text{if } s = s'. \end{cases} \tag{6}$$

Following Kakade & Langford (2002), we calculate the difference between $J_{\pi_{\text{post}}^\delta}$ and $J_{\pi_{\text{post}}^*}$.

$$J_{\pi_{\text{post}}^\delta} - J_{\pi_{\text{post}}^*} = \frac{1}{1-\gamma} \mathbb{E}_{s \sim d_\mu^{\pi_{\text{post}}^\delta}} \mathbb{E}_{a \sim \pi_{\text{post}}^\delta} \left[ A^{\pi_{\text{post}}^*}(s, a) \right], \tag{7}$$

where $d_\mu^{\pi_{\text{post}}^\delta}$ is the discounted steady-state distribution of the MDP under policy $\pi_{\text{post}}^\delta$ and start state distribution $\mu$, and $A^{\pi_{\text{post}}^*}(s, a)$ is the advantage function of policy $\pi_{\text{post}}^*$. We can further simplify equation 7 considering the deterministic nature of $\pi_{\text{post}}^*$ and $\pi_{\text{post}}^\delta$ as:

$$J_{\pi_{\text{post}}^\delta} - J_{\pi_{\text{post}}^*} = \frac{1}{1-\gamma} \mathbb{E}_{s \sim d_\mu^{\pi_{\text{post}}^\delta}} \left[ Q^{\pi_{\text{post}}^*}(s, \pi_{\text{post}}^\delta(s)) - Q^{\pi_{\text{post}}^*}(s, \pi_{\text{post}}^*(s)) \right] \tag{8}$$

Note that the two policies differ only at a state $s'$. This allows us to further simplify as:

$$= \frac{d_\mu^{\pi_{\text{post}}^\delta}(s')}{1-\gamma} \left[ Q^{\pi_{\text{post}}^*}(s', \pi_{\text{post}}^\delta(s')) - Q^{\pi_{\text{post}}^*}(s', \pi_{\text{post}}^*(s')) \right] \geq \frac{d_\mu^{\pi_{\text{post}}^\delta}(s')}{1-\gamma} (-\delta) \geq -\frac{\delta}{1-\gamma} \tag{9}$$

The inequalities follow from the agent-labeler disagreement definition and $0 \leq d_\mu^{\pi_{\text{post}}^\delta}(s) \leq 1$. $\qquad \square$

Next, we show how this proof extends to multiple agent-labeler disagreements.

**Corollary 4.4.** *When there are multiple agent-labeler disagreements, $\{\delta_1, \delta_2, \cdots, \delta_k\}$, the Theorem 4.3 can be extended as: $J_{\pi_{post}^\delta} \geq J_{\pi_{post}^*} - \frac{\max\{\delta_1, \delta_2, \cdots, \delta_k\}}{1-\gamma}$.*

*Proof.* The proof follows similar reasoning as the single disagreement case, considering the worst-case scenario where the maximum disagreement dominates the performance loss. $\qquad \square$

*Remark* 4.5 (Beyond the noiseless, deterministic setting). The noiseless-preference and deterministic-policy assumptions correspond to the $\alpha \to \infty$ limit of the belief-based model. For finite $\alpha$, the Luce choice axiom (Luce, 1959) instead yields a stochastic softmax policy, and the same Performance Difference Lemma argument gives a bound that depends additionally on the maximum advantage magnitude at the disagreement state; it recovers Theorem 4.3 as $\alpha \to \infty$. We give this generalized bound and its proof in Appendix A. A more substantial relaxation is to move beyond the tabular setting: with function approximation, a belief perturbation at one state can shift the policy elsewhere through shared parameters, so our localized single-state argument no longer applies directly. We leave a bound for this setting to future work, and study it empirically in Section 5.

## 5 Empirical Study of Impact of Agent-Labeler Disagreements on Policy Learning

In the prior section, we bounded the effects of agent-labeler disagreement for simple tabular policies without function approximation or generalization. In this section, we aim to empirically study this phenomenon for a broader class of policy representations that can exhibit generalization, as well as limit the set of representable policies. As noted earlier, when infinite noiseless preferences are given, then the normative ideal is for the labeler to give preferences with respect to an optimal policy, as long as the agent can represent and execute an optimal policy. However, if maximum agent performance is limited in some manner—for example, due to a weak function approximator, regularized learning rule, or noisy actuators that don't faithfully execute the commanded action—then the normative ideal is to give preferences with respect to the best achievable policy.

Table 1: The effect of mismatch in belief in the agent capabilities (rows) and the agent's actual capability (columns). Highest returns generally correspond to the most aligned beliefs, shown along the diagonal. The error intervals show the 95% confidence interval.

| $\epsilon$ noise on agent capability $\rightarrow$ | 0.0 | 0.1 | 0.3 | 0.5 |
|---|---|---|---|---|
| $\epsilon'$ noise on labeler's belief $\downarrow$ | | | | |
| 0.0 | $\mathbf{7.92 \pm 0.05}$ | $\mathbf{3.11 \pm 0.15}$ | $-4.07 \pm 0.21$ | $-8.45 \pm 0.26$ |
| 0.1 | $3.49 \pm 0.07$ | $1.47 \pm 0.17$ | $-4.92 \pm 0.22$ | $-10.91 \pm 0.32$ |
| 0.3 | $1.71 \pm 0.05$ | $-0.88 \pm 0.14$ | $\mathbf{-3.70 \pm 0.19}$ | $-8.18 \pm 0.26$ |
| 0.5 | $3.47 \pm 0.06$ | $-0.32 \pm 0.11$ | $-4.14 \pm 0.22$ | $\mathbf{-7.36 \pm 0.25}$ |

To study the impact of agent-labeler disagreement in a more realistic setting in which the agent is limited, we design an experiment using synthetic data in a gridworld domain. The agent can only represent epsilon-greedy policies for a particular value of epsilon. This policy noise is intended to serve as a stand-in for the various types of performance-capping limitations discussed above, and has the advantage of being easy to systematically manipulate. We model the labeler's beliefs very simply as a belief over epsilon; they assume an optimal policy within the representable epsilon-greedy class. Thus, agent-labeler disagreement arises due to incorrect beliefs about the value of epsilon. We hypothesize that performance of the post-RLHF policy will degrade as the labeler's belief over epsilon diverges from the true epsilon.

We first collect 100 random trajectories (enough to guarantee a nearly optimal-within-class post-RLHF policy) of the agent traversing the gridworld environment that terminate either in the goal state or in the cliff state. We denote $\epsilon$ as the $\epsilon$-greedy noise impacting the post-RLHF policy of the agent and $\epsilon'$ as the $\epsilon$-greedy noise assumed by the labeler and determining the labeler's preferences. The values of $\epsilon$ and $\epsilon'$ may be different. We denote the labeler's belief about agent's capability $Q^{\pi_{\text{belief}}^{\epsilon'}}(s,a)$. We denote the post-RLHF agent capability as $Q^{\pi_{\text{post}}^{\epsilon}}(s,a)$. We vary the values of $\epsilon$ and $\epsilon'$ to show the impact of agent-labeler disagreements on the post-RLHF policy's expected return.

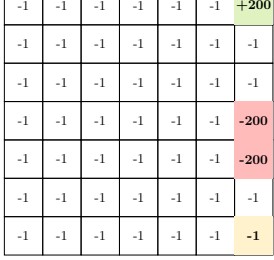

Figure 3: 7x7 GridWorld with start state (in yellow), two terminal cliff states (in red) and one terminal goal state (in green). In each cell, we mark the reward incurred for reaching the state.

We use $Q^{\pi_{\text{belief}}^{\epsilon'}}(s,a)$ to generate synthetic preference data using the belief-based preference model shown in Equation 3. We use these preferences to train the agent using Contrastive Preference Learning (CPL) (Hejna et al., 2024), a scalable algorithm for regret-based RLHF.

Table 1 shows the average returns of the post-RLHF policies. The highest return generally comes when the $\epsilon'$-greedy noise assumed by labeler is closest to the $\epsilon$-greedy noise impacting the agent's policy. We see that for lower values of epsilon ($\epsilon = 0 \,\&\, 0.1$), preferences given under the optimal advantage function ($\epsilon' = 0$) provide the highest returns. However, this strategy becomes a hindrance for effective RLHF as the action noise increases, as seen when $\epsilon = 0.3 \,\&\, 0.5$. This is due to the fact that the post-RLHF policy learns to walk near the dangerous cliffs, often falling off, since the labeler assumed too low of a noise level. Walking near the cliffs is only optimal when the agent has a very low epsilon value. These results serve as a proof of concept for how agent-labeler disagreements can arise in practice and how they can be exacerbated by agent limitations that may be unknown to the labeler.

## 6 Agent-Labeler Disagreements in Human Preference Collection

Now that we have established that agent-labeler disagreements pose a problem in RLHF, in this section, we demonstrate that human beliefs about agent capabilities do, in fact, significantly affect the preferences given by human labelers. This is done via a human study in which priming effects are used to change participants' beliefs about the agent's capabilities. Additionally, this result indicates that it is possible in theory to change human preferences to be better aligned with agent capabilities.

**Domain:** We choose a self-driving car domain, simulated using the CARLA driving simulator (Dosovitskiy et al., 2017) to collect trajectories to use for human preference collection. Since the rules and trade-offs involved

in driving are something that many people already have a strong intuition for, the choice of self-driving car setup allows us to keep the filtering of the participants minimal.

**Priming:** Before beginning data collection, we randomly assign participants to one of three conditions: (1) no priming (control condition), (2) safe priming (confidence increasing condition), and (3) unsafe priming (confidence decreasing condition). Both safe and unsafe priming videos start from the same starting state to avoid unintended biasing effects. Participants in the safe priming group are shown a video where the car obeys all traffic laws, avoids obstacles, and successfully overtakes a slower car, whereas the participants in the unsafe priming group are shown a video where the car drifts across lanes and crashes into obstacles.

**Preference elicitation:** Participants are asked to provide preferences over 15 randomly chosen pairs of video data. All data pairs on which preferences are collected show legal driving. Within each pair, the car is shown to be driving on the same path for the same amount of time. In each pair, one video shows faster and more time saving behavior, which if executed imperfectly (during or after the trajectory) could lead to failure states, such as hitting a pedestrian, another car, or breaking traffic laws; the other shows more safety-conscious behavior that requires more time to execute, but for which it is easier to avoid failure states. Each pair of trajectories is designed to make the participants consider both: (a) how a learning agent might behave in the same states shown in the trajectories and (b) what the agent might do after the end of each trajectory.

All data pairs are randomly shuffled, and the order in which each pair of data is shown is also randomly shuffled to avoid ordering bias during preference collection. In each question, both videos in each pair are presented next to each other.

**Instructions for the participants:** Participants in the control group are given a standard preference collection instruction explaining that the objective of the survey is to improve a self-driving car using their preferences over behavioral data. We chose to frame this language for this group to indicate that their preferences were not meant to be a rating of the trajectories themselves, but rather a learning signal for improving a self driving car's policy. This was designed to lead them to reflect less on the trajectories themselves and more on their beliefs about how a learning agent might behave in similar or future states. The participants in the safe and unsafe priming groups are given a similar instruction *with an addition of a video showcasing the car's driving behaviors.* To encourage the participants to think deeply about the car's capabilities, they are then asked about which skills they think would be useful for the car to learn to become a more effective driver.

**Post-processing of collected data:** To maintain a high standard while finalizing the data, we construct a participant filtering step that ensures that participants are paying attention when providing preferences. First, we show a pair of videos: one in which the car drives normally and one in which the car crashes into a guardrail while making a turn. Additionally, we show a pair of videos in which one car drives normally and the other runs a red light. We remove all preference data from participants who do not strongly prefer the attention check videos that show good driving, while also reporting that they were extremely confident.

In total, we collected data from 259 participants. The data filtering and quality checks allowed 146 (46 safe priming, 50 unsafe priming, 50 no priming) participants' data to be used for analysis. We further balanced the data in each group to avoid unintended impacts on the statistical analysis by randomly removing 4 entries from both the unsafe and no priming groups. The final dataset contains 46 participants in each of the three groups.

To the best of our knowledge, there is not a standard nonparametric statistical test that can test for statistical significance when there are: (1) more than two groups; (2) for which the data is Likert; (3) and for which each participant gives repeated measures data. Instead, we used the average of each participant's responses, rather than individual question responses, to create independent data that is amenable to standard statistical tests. To reduce noise due to participant confusion, uncertainty, or misunderstandings, we used only those responses where participants reported extremely high confidence in their answers.

We briefly justify these analysis choices. *Averaging repeated responses.* A natural alternative is a mixed-effects ordinal regression, such as a cumulative link mixed model (CLMM), which can directly model repeated ordinal measures. However, CLMMs rely on the proportional-odds assumption—that priming shifts responses

uniformly across all thresholds of the Likert scale. In our setting priming may plausibly affect the extremes of the scale differently from its middle, and we did not have strong grounds to assume proportional odds holds; moreover, with 46 participants per group after filtering, the effective sample size is small for stably estimating the random-effects variance components of a CLMM. Averaging each participant's responses instead yields a single independent summary statistic per participant, which enables the Kruskal–Wallis test (Kruskal & Wallis, 1952)—a well-established nonparametric test with minimal distributional assumptions. This discards some within-participant information, but we preferred a test whose assumptions we could be confident about over a more powerful test whose assumptions may be violated. *Retaining only high-confidence responses.* In our experience with online platforms such as Prolific, responses given with low self-reported confidence frequently reflect participant disengagement or rushed responding rather than genuine uncertainty about the preference—an effect compounded here, where participants must watch and compare two videos before answering. The confidence filter thus complements the attention-check filtering described above as a quality-control measure. *Robustness.* Our analysis is conservative by design: the Bonferroni correction used in the Dunn's test is among the most stringent multiple-comparison corrections available, and the Kruskal–Wallis test makes minimal distributional assumptions. We still observe significance at the 5% level (Table 2, $p = 0.015$ between safe and unsafe priming), and a less conservative analysis would be expected to only strengthen these results.

**Qualitative results:**

Immediately after priming, we asked participants to provide free-text feedback directed specifically at the car's capabilities—namely, which skills they think would be useful for the car to learn to become a more effective driver. Because this question is directed at the agent's capabilities rather than driving risk in general, the responses serve as a probe of participants' beliefs about the agent. To quantify these responses, we used GPT-4 (OpenAI, 2024) to conduct a sentiment analysis for each response and provide a 1–10 star rating, using the following prompt:

> *"Analyze the following feedback about a car's driving skills and rate its safety from 1 to 10 stars. Are there any skills this car need to learn to be a more effective driver? Only consider the presence of any safety concerns and suggestions for improvement with small consideration for factors unrelated to safety: Feedback: [PARTICIPANT'S FEEDBACK] Provide a structured analysis with reasons for the assigned score."*

GPT-4 was the strongest model available at the time our study was conducted. We chose an LLM over human sentiment annotators for greater consistency, as LLM-based annotations have been shown to achieve higher internal consistency than human annotators for sentiment analysis tasks (Bojić et al., 2025; Zhang et al., 2024); rating the safety sentiment of short free-text responses on a numerical scale is straightforward relative to the benchmarks studied in that work.

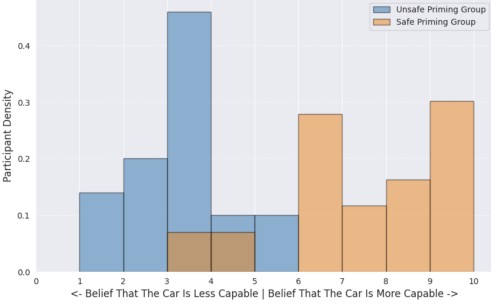

Figure 4: Sentiment analysis on participants' written responses about the car's capabilities after priming, evaluated on a 1–10 scale by GPT-4. Histogram bars for both groups are plotted from 0, not stacked.

Figure 4 shows that immediately after priming, participants primed with the unsafe priming had lower sentiment scores regarding the car's capabilities than the participants in the safe priming group, suggesting

that priming had an impact on their beliefs about the agent specifically. This establishes the first link in the causal chain we aim to demonstrate—that priming shifts beliefs about agent capabilities. The Kruskal–Wallis and Dunn–Bonferroni analysis of the Likert preference data below establishes the second link—that these shifted beliefs in turn affect labeling behavior.

**Quantitative results:**

Table 2: P-values for Dunn-Bonferroni for responses given with extremely strong confidence. P-values below 0.05 indicate statistical significance at the 5% level.

|  | Unsafe Priming | No Priming | Safe Priming |
|---|---|---|---|
| Unsafe Priming | 1.000 | 0.229 | 0.015 |
| No Priming | 0.229 | 1.000 | 0.847 |
| Safe Priming | 0.015 | 0.847 | 1.000 |

Next, we analyzed the Likert data associated with the preferences of each condition group. We used the non-parametric Kruskal-Wallis test for statistical analysis due to the ordinal nature of the data Kruskal & Wallis (1952). The Kruskal-Wallis test resulted in a p-value of 0.033, indicating a statistically significant difference between at least two priming groups at the 5% level. In order to detect which of the two priming groups differ, we used the Dunn's test with Bonferroni correction Dunn (1961). The results of the Dunn-Bonferroni test are shown in Table 2 and show statistically significant ($p = 0.015$) differences between safe and unsafe priming groups at the 5% level. Cliff's delta (d = 0.31) between unsafe vs safe priming shows that participants primed with unsafe priming were influenced toward more safe responses at a higher rate than participants primed with safe priming with a small effect size Cliff (1993). Participants in the no priming group did not exhibit statistically significant differences in their responses compared to either the unsafe or safe priming groups, as shown in Table 2. These results suggest that human preferences are influenced by their beliefs about agent capabilities. To mitigate this bias, practitioners should actively attempt to strengthen agent-labeler agreement during data collection.

## 6.1 Recommendations For Researchers

Our results show that having good agent-labeler agreement is important for effective policy learning in regret-based RLHF. While it is difficult to reason about an agent's post-training performance prior to training, our results suggest potential best practices for practitioners:

**(1) Inform labelers directly about known limitations:** Practitioners may choose to inform labelers about agent limitations prior to preference collections. For example, this may include known data or training limitations, non-obvious restrictions on agent capabilities (e.g. an unusually limited turning radius on a vehicle), or assessments of projected agent strengths and weaknesses. However, the success of such an approach depends heavily on the ability of labelers to make sense of such information, which may require varying levels of technical expertise.

**(2) Online preference collection with intermittent priming:** Practitioners may choose to use online preference collection interleaved with RLHF training on the collected preferences. Here, labelers' beliefs about the agent's capability may be continuously updated and aligned via priming using data from the agent's most current policy. This approach may also demonstrate to labelers how their preferences influence policy learning.

Both recommendations reduce to the same mechanism: exposing labelers to the agent's actual behavior before they provide preferences. Our human study is a proof of concept that this is feasible and effective—a single priming video shifted preferences significantly (Table 2)—and the online loop of recommendation (2) adds only one step to a standard pipeline: generating demonstrations, which requires merely running the current policy. These rollouts need not come from a post-RLHF policy; the agent's policy at any training stage is informative about its current capabilities. What constitutes an informative demonstration is domain-specific (e.g., for a language-model agent), but the mechanism of calibrating beliefs through exposure to behavior is general.

**Truthful calibration versus manipulation.** Because priming can shift preferences, calibration must be distinguished from manipulation: the difference is whether the demonstrations are representative of the agent's actual on-policy behavior. Truthful calibration shows unfiltered rollouts reflecting real capabilities and limitations; the same interface could instead curate a misleading picture (e.g., hiding failure modes) to

steer labels toward a preferred outcome. We therefore recommend sampling demonstrations representatively from the current policy, as non-representative behavior undermines both labeler autonomy and the reliability of the reward signal.

## 7 Conclusion

Current methods of preference collection do not account for participants' beliefs about the capabilities of the agents learning from their preferences. We show that the preferences collected under agent-labeler disagreements can lead to suboptimal policies both theoretically and empirically and make preliminary suggestions on how to minimize such disagreements. Future work may include developing improved priming strategies to ensure human beliefs are as accurate as possible, as well as algorithmic advances that mitigate the impact of incorrect beliefs on RLHF.

**Limitations and open directions.** Each part of our analysis suggests a natural next step. Theorem 4.3 is proved in a clean tabular setting with noiseless preferences; Appendix A relaxes the noiseless assumption, and extending it to function approximation—where a belief perturbation can propagate across states through shared parameters—is a promising direction for sharper guarantees. Empirically, our synthetic experiments isolate the effect under controlled conditions and our human study shows that priming shifts both beliefs and preferences; validating gains in a full RLHF pipeline driven end-to-end by human labels, which would require a substantially larger preference budget, is an exciting avenue for follow-up work.

### Acknowledgments

This work has taken place in the Safe, Correct, and Aligned Learning and Robotics Lab (SCALAR) at The University of Massachusetts Amherst. SCALAR research is supported in part by the NSF (IIS-2437426) and Open Philanthropy.

Scott Niekum holds concurrent appointments as an Associate Professor at the University of Massachusetts Amherst and as an Amazon Scholar. This paper describes work performed at the University of Massachusetts Amherst and is not associated with Amazon.

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

## A  Proofs

Proof of the corollary 4.4:

*Proof.* Theorem 4.3 states that the error bound on the expected return of a post-RLHF policy learned with preferences given under a single agent-labeler agreement at state $(s', a')$ of magnitude $\delta$ is:

$$J_{\pi_{\text{post}}^{\delta}} \geq J_{\pi_{\text{post}}^{*}} - \frac{\delta}{1 - \gamma}.$$

This result can be interpreted as, in the worst case, the agent takes a sub-optimal action of magnitude $\delta$ at state $s'$ from the start of the episode until the end of horizon. Consider the case of multiple disagreements where we have state-action pair $(s'', a'')$ where we have disagreement of magnitude $\max\{\delta_1, \delta_2, ..., \delta_k\}$ . In the worst case, the agent will be stuck at state $(s'', a'')$ from the beginning of the episode until of the horizon. This results in following bound on expected returns:

$$J_{\pi_{\text{post}}^{\delta}} \geq J_{\pi_{\text{post}}^{*}} - \frac{\max\{\delta_1, \delta_2, ..., \delta_k\}}{1 - \gamma}.$$

$\square$

### A.1  A visitation-weighted bound for multiple disagreements

The bound in Corollary 4.4 is deliberately assumption-free: it replaces every disagreement by the largest one and every visitation probability by its upper bound of 1. When the visitation distribution of the perturbed policy is available, the same argument yields a tighter, visitation-weighted bound that makes explicit how disagreements at different states interact through the MDP.

**Proposition A.1.** *Consider the setting of Theorem 4.3, but with agent-labeler disagreements of magnitudes $\delta_1, \delta_2, \ldots, \delta_k$ at distinct state-action pairs $(s'_1, a'_1), \ldots, (s'_k, a'_k)$, each large enough to flip the post-RLHF action at its state. Let $\pi_{post}^{\delta}$ be the resulting post-RLHF policy and $d_{\mu}^{\pi_{post}^{\delta}}$ its discounted state-visitation distribution. Then*

$$J_{\pi_{post}^{\delta}} \geq J_{\pi_{post}^{*}} - \frac{1}{1 - \gamma} \sum_{i=1}^{k} d_{\mu}^{\pi_{post}^{\delta}}(s'_i) \, \delta_i. \tag{10}$$

*Proof.* As in the proof of Theorem 4.3, when RLHF respects the preferences it returns the deterministic policy $\pi_{\text{post}}^{\delta}(s) = \arg\max_a Q^{\pi_{\text{belief}}}(s, a)$, which now differs from $\pi_{\text{post}}^{*}$ at exactly the $k$ states $s'_1, \ldots, s'_k$ where a disagreement flips the preferred action. Applying the Performance Difference Lemma (Kakade & Langford, 2002) and using that both policies are deterministic,

$$J_{\pi_{\text{post}}^{\delta}} - J_{\pi_{\text{post}}^{*}} = \frac{1}{1 - \gamma} \mathbb{E}_{s \sim d_{\mu}^{\pi_{\text{post}}^{\delta}}} \left[ Q^{\pi_{\text{post}}^{*}}\big(s, \pi_{\text{post}}^{\delta}(s)\big) - Q^{\pi_{\text{post}}^{*}}\big(s, \pi_{\text{post}}^{*}(s)\big) \right]$$

$$= \frac{1}{1 - \gamma} \sum_{i=1}^{k} d_{\mu}^{\pi_{\text{post}}^{\delta}}(s'_i) \left[ Q^{\pi_{\text{post}}^{*}}\big(s'_i, a'_i\big) - Q^{\pi_{\text{post}}^{*}}\big(s'_i, \pi_{\text{post}}^{*}(s'_i)\big) \right], \tag{11}$$

where the sum is over only the $k$ states at which the policies differ, since the bracketed term is zero elsewhere. By the definition of agent-labeler disagreement, each bracketed term is at least $-\delta_i$, giving

$$J_{\pi_{\text{post}}^{\delta}} - J_{\pi_{\text{post}}^{*}} \geq -\frac{1}{1 - \gamma} \sum_{i=1}^{k} d_{\mu}^{\pi_{\text{post}}^{\delta}}(s'_i) \, \delta_i.$$

$\square$

This form makes the interaction between disagreements transparent: a disagreement at a rarely-visited state $s_i'$ (small $d_\mu^{\pi_{\mathrm{post}}^\delta}(s_i')$) contributes proportionally little to the performance loss, whereas a disagreement on the agent's main path is weighted heavily. Corollary 4.4 is recovered by bounding each $d_\mu^{\pi_{\mathrm{post}}^\delta}(s_i') \leq 1$ and each $\delta_i \leq \max_j \delta_j$, together with $\sum_i d_\mu^{\pi_{\mathrm{post}}^\delta}(s_i') \leq 1$.

## A.2 A generalized bound for noisy preferences

Theorem 4.3 assumes noiseless preferences ($\alpha \to \infty$), which yield a deterministic post-RLHF policy. We now relax this assumption and derive a bound for finite $\alpha$, where preferences are noisy and the post-RLHF policy is stochastic. As noted in Remark 4.5, this bound recovers Theorem 4.3 in the $\alpha \to \infty$ limit. We emphasize that the quantities below (in particular the advantage magnitude $A_{\max}$) serve a purely analytical role and are not intended to be computed in practice.

**Proposition A.2.** *Consider the setting of Theorem 4.3 with a single agent-labeler disagreement of magnitude $\delta$ at $(s', a')$, but with a finite inverse-temperature $\alpha$ in the belief-based preference model (Eqn. 3). Suppose RLHF returns the softmax policy consistent with the resulting pairwise preferences (Luce choice axiom (Luce, 1959)), $\pi(a \mid s) \propto \exp\big(\alpha \, Q^{\pi_{belief}}(s, a)\big)$, and let $\pi_{post}^\delta$ denote this policy under the labeler's belief. Let $p = \pi_{post}^*(a' \mid s')$ be the probability the ideal policy assigns to $a'$ at $s'$, and let $A_{\max}(s') = \max_a \big|A^{\pi_{post}}(s', a)\big|$. Then*

$$J_{\pi_{post}^\delta} \geq J_{\pi_{post}^*} - \frac{2 \, d_\mu^{\pi_{\mathrm{post}}^\delta}(s') \, A_{\max}(s') \, p(1-p)\big(e^{\alpha\delta} - 1\big)}{(1-\gamma)\big(1 + p(e^{\alpha\delta} - 1)\big)}. \tag{12}$$

*Proof.* Under the Luce choice axiom, a disagreement of magnitude $\delta$ raises the belief-value of $a'$ at $s'$ by $\delta$ relative to the ideal belief, i.e. $Q^{\pi_{\mathrm{belief}}}(s', a') = Q^{\pi_{\mathrm{belief}}^*}(s', a') + \delta$, while leaving the other belief-values at $s'$ unchanged. The resulting softmax probability the perturbed policy places on $a'$ is

$$\pi_{\mathrm{post}}^\delta(a' \mid s') = \frac{p \, e^{\alpha\delta}}{1 + p\big(e^{\alpha\delta} - 1\big)}, \tag{13}$$

where $p = \pi_{\mathrm{post}}^*(a' \mid s')$ is the corresponding probability under the ideal belief. Hence the increase in probability mass on $a'$ is

$$\Delta p := \pi_{\mathrm{post}}^\delta(a' \mid s') - p = \frac{p(1-p)\big(e^{\alpha\delta} - 1\big)}{1 + p\big(e^{\alpha\delta} - 1\big)} \geq 0, \tag{14}$$

and this mass is removed from the other actions at $s'$. The two policies differ only at $s'$, so by the Performance Difference Lemma (Kakade & Langford, 2002),

$$J_{\pi_{\mathrm{post}}^\delta} - J_{\pi_{\mathrm{post}}^*} = \frac{d_\mu^{\pi_{\mathrm{post}}^\delta}(s')}{1 - \gamma} \, \mathbb{E}_{a \sim \pi_{\mathrm{post}}^\delta(\cdot|s')}\Big[A^{\pi_{\mathrm{post}}^*}(s', a)\Big], \tag{15}$$

since $\mathbb{E}_{a \sim \pi_{\mathrm{post}}^*}[A^{\pi_{\mathrm{post}}^*}(s', a)] = 0$ and the advantage term vanishes at every state other than $s'$. Because $\pi_{\mathrm{post}}^\delta$ and $\pi_{\mathrm{post}}^*$ agree at $s'$ except for the mass $\Delta p$ shifted onto $a'$ from other actions, the expected advantage differs from zero by at most $2A_{\max}(s') \, \Delta p$ in magnitude:

$$\left| \mathbb{E}_{a \sim \pi_{\mathrm{post}}^\delta(\cdot|s')}\Big[A^{\pi_{\mathrm{post}}^*}(s', a)\Big] \right| \leq 2 \, A_{\max}(s') \, \Delta p. \tag{16}$$

Combining this with the Performance Difference Lemma and substituting $\Delta p$ from Eqn. equation 14 gives

$$J_{\pi_{\mathrm{post}}^\delta} - J_{\pi_{\mathrm{post}}^*} \geq -\frac{2 \, d_\mu^{\pi_{\mathrm{post}}^\delta}(s') \, A_{\max}(s') \, p(1-p)\big(e^{\alpha\delta} - 1\big)}{(1-\gamma)\big(1 + p(e^{\alpha\delta} - 1)\big)}.$$

$\square$

As $\alpha \to \infty$, the factor $\Delta p \to 1 - p$, so $\pi_{\text{post}}^{\delta}(a' \mid s') \to 1$ and the softmax policy collapses to the deterministic action switch of Theorem 4.3; the analysis then reduces to that theorem, and under its assumptions the bound simplifies to $\delta/(1-\gamma)$, which avoids the dependence on $A_{\max}(s')$. We note that the generalized bound above is intended to give qualitative intuition for how performance degrades under relaxed assumptions rather than to be a tight characterization.

## B  Additional Details About Human Study

In this section, we provide more details about our human study and the demographics of participants.

### B.1  Design Decisions

We chose self-driving cars as the domain on which to collect our preference dataset because the rules and trade-offs of driving are something people understand without high level knowledge about any language or formal higher education and allowed for minimal filtering of participants. We conducted these preference collection experiments with prior IRB approval.

### B.2  Data Collection UI

Figure 5 shows the format of the survey that participants saw when giving their responses.

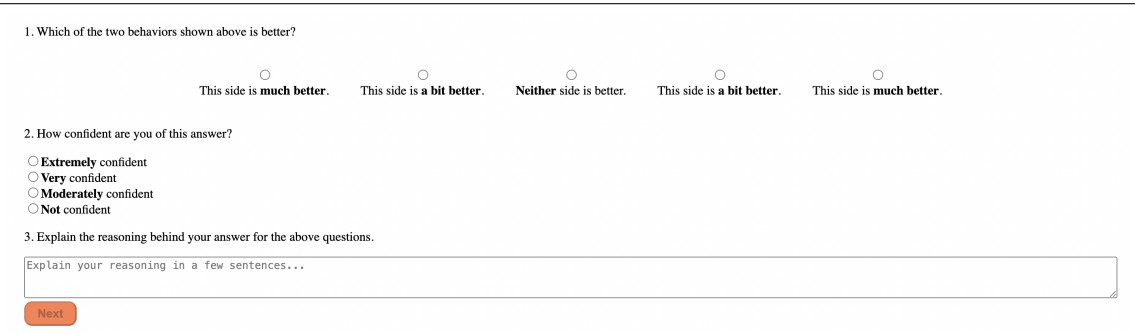

Figure 5: The format of questions asked to participants in the survey. In order to avoid order biasing, we avoided terms such as "first/second", "left/right", or arrows and opted to use labels to indicate which side they prefer more. Subjects are also asked to report confidence in their response and a reasoning for their response. Subjects are required to respond to all three questions for each preference pair before they are allowed to move to the next pair in order to avoid confusion.

### B.3  Example of a Pair of Trajectories Shown to Human Labelers

All trajectories shown to participants show legal driving. Each pair of trajectories show the car driving along the same route for the same amount of time so participants can easily distinguish the tradeoff between safety and time savings. For example, one pair of scenarios shows a legal overtake of a slow car in a single lane road, as shown in Figure 6. While it is legal and safe to perform the overtake, doing so requires trust in the car's abilities to successfully return back to its lane without any accidental swerving or any other mishaps. The pair shows one trajectory where the car does not attempt this overtake and instead patiently stays behind the slow car, as shown in Figure 6a; the other shows the car overtaking successfully and getting further along the road in the same amount of time, as shown in Figure 6b. This allows room for the subject to interpret the quality of the car's driving for the overtake and be subjective in their belief in the car's abilities.

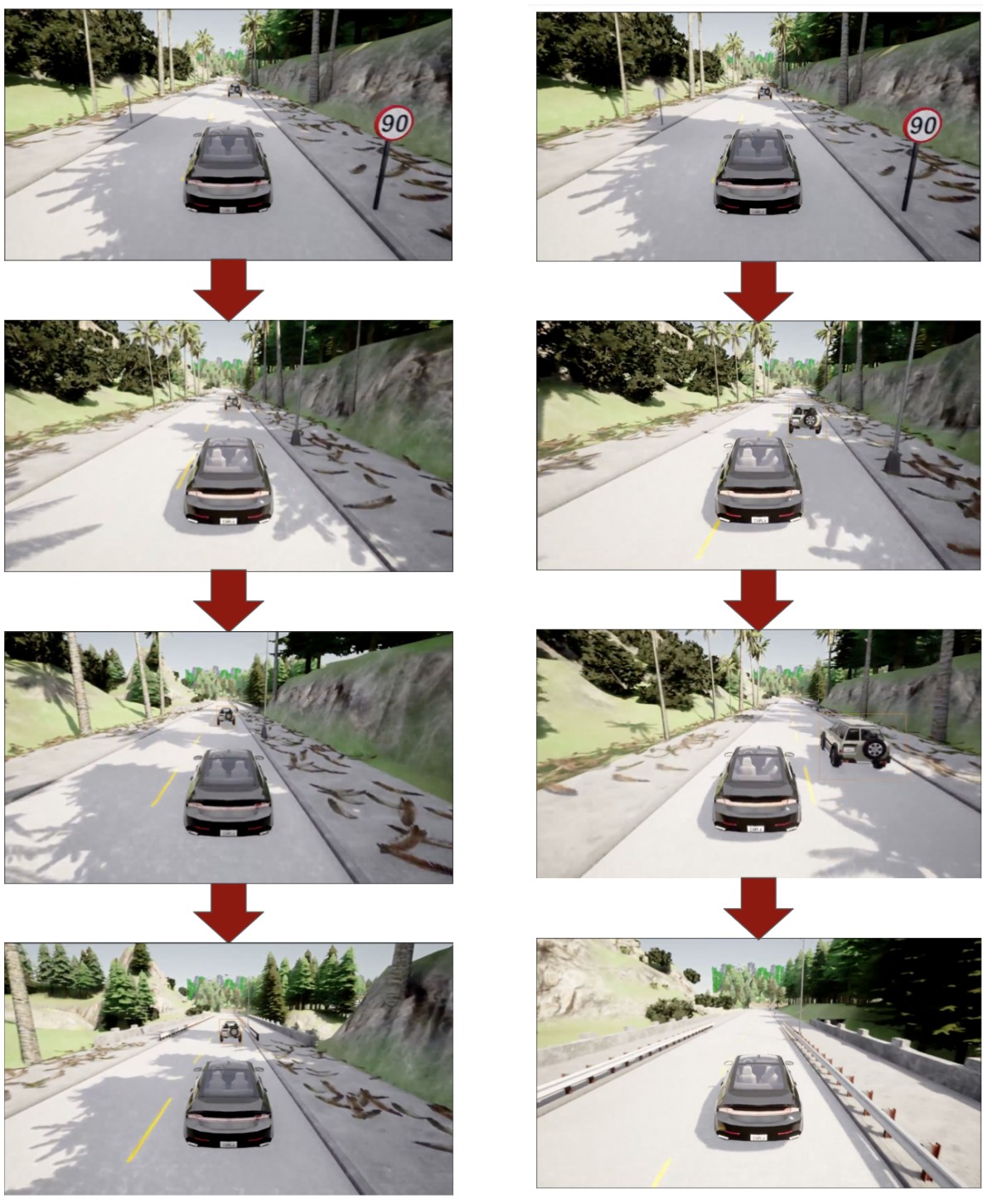

(a) An example of a trajectory showing safety-conscious driving behavior. Here, the car chooses to stay behind the slower driver and finishes the trajectory near the start of the bridge.

(b) An example of a trajectory showing time-saving driving behavior. Here, the car chooses to legally overtake the slow driver and finishes the trajectory near the end of the bridge.

Figure 6: An example pair of trajectories as shown to human labelers. Both trajectories show the car driving along the same path for the same amount of time.

### B.4 Demographics

Figure 7a shows the distribution of driver experience among the participants of our survey and Figure 7b shows participant's willingness to ride in a self driving car. While we did not have specific filters for years of driving experience, most of the participants report a significant number of years of driving experience, meaning that they are well versed in the domain. A large portion of participants from all three priming groups reported that they would be willing to ride in a self driving car. However, out of all participants who reported that they would be unwilling to ride in a self driving car, the largest group of participants was the unsafe priming group.

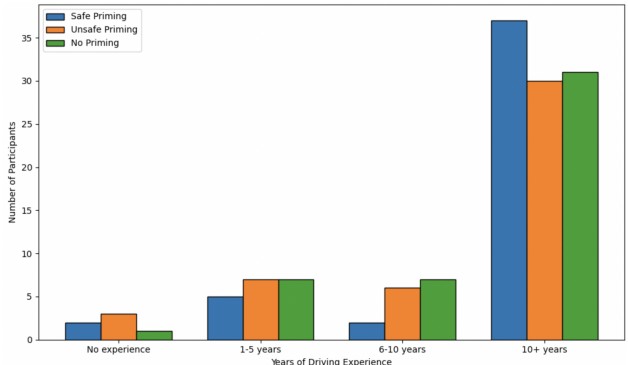 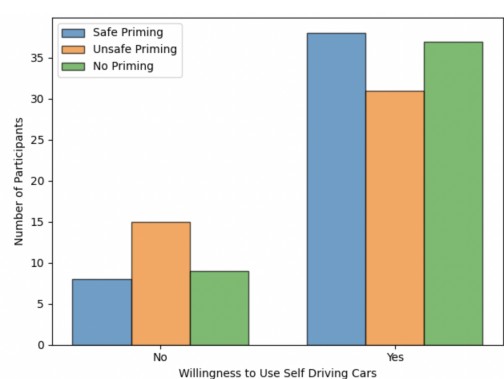

(a) Participant's driving years of experience.      (b) Participant's willingness to use self-driving cars.

Figure 7: Demographics of human labelers

## C  GridWorld Experimental Details

In this section, we provide additional details about the GridWorld domain and the hyperparameters used in the learning algorithm. For this analysis, we used the following python libraries: numpy Harris et al. (2020), math and random from the standard python library Python Software Foundation (2023), and scipyVirtanen et al. (2020).

### C.1  Gridworld Domain

The gridworld domain consists of a 7x7 grid of cells. We number the coordinates from 0 to 6. The agent always starts in the cell numbered (6,6). From there, the agent may move in any of the four cardinal directions. Any movement by the agent that results in the agent moving outside the boundaries of the environment results in the agent staying in its original state with an additional reward of -1. The states $(0, 6)$, $(3, 6)$, and $(4, 6)$ are terminal states and result in a reward of $+200$, $-200$, and $-200$ respectively. The episode terminates when the agent either reaches one of the terminal states or when the number of states that the agent has visited reaches 1,000.

### C.2  Hyperparameters

Here, we will discuss the hyperparameters used in the gridworld experiment for the Contrastive Preference learning algorithm:

| Hyperparameter | Value |
| --- | --- |
| Discount Factor | 0.7 |
| Regularization ($\gamma$) | 0.01 |
| Learning Rate | 0.5 |
| Temperature parameter ($\alpha$) | 10 |
| Number of random seeds | 20 |
| Epochs | 20 |

Table 3: Hyperparameters used in the gridworld experiment

## D    Disclosure about LLM usage

We used LLM to format the content of the paper and to make any stylistic changes to the paper.

