# OpenReview forum: "A Descriptive and Normative Theory of Human Beliefs in RLHF"
_TMLR — Accepted by TMLR_

### Review · Reviewer_5Bvg · 2026-03-31

**Summary Of Contributions:**

This paper is about modeling the human preferences in reinforcement learning with human feedback. This work proposes that human beliefs about the capabilities of the agents play a key role in preference generation. It proposes a preference model that incorporate human beliefs and provide a normative theory that bounds the error on the final learned policy based on the mismatch between the human’s beliefs and an idealized set of beliefs. It shows that human beliefs about agent capabilities can significantly affect preferences through interventions. The paper also shows that assuming agent’s optimality can lead to suboptimal behavior for generating preferences.

**Audience:**

Yes

**Audience Explanation:**

The paper contributes to the field of AI alignment, specifically human preference modeling in RLHF. Taking into account the belief about the agent’s capabilities is an new idea given that RLHF usually treats preference labels as if they reflect reward alone. This paper gave a framework for this issue and may benefit those who are interested in learning about preference modeling in RLHF.

**Broader Impact Concerns:**

No concerns.

**Claims And Evidence:**

No

**Claims Explanation:**

- Theorem 4.3 gives a bound on expected return under a very narrow setting: single-transition comparisons, noiseless preferences, tabular deterministic policies, and an RLHF procedure that exactly respects all preferences. It seems like these assumptions are much stronger than the practical RLHF settings the paper is trying to influence, so the theorem serves more as an existence proof than as strong support for practical recommendations. The extension to multiple disagreements is especially weak when the corollary simply takes the maximum disagreement and appeals to a worse-case argument.  It doesn’t discuss how multiple disagreements may interact in the MDP setup.
- The notion of the ‘ideal’ belief is somewhat underdeveloped.
- Section 5 only uses epsilon-greedy noise, it is not clear how well the conclusions transfer to more realistic RLHF settings involving function approximation, distribution shift, or partial observability.
- The human study uses the average of each participant’s responses, but I think there exists statistical tests that can handle this e.g. mixed-effects ordinal regression. The study also only keeps the responses where the participants show high confidence in their answers. This may discard important information.
- It is not clear why the paper uses GPT-4 to conduct a sentiment analysis on the textual feedback and the paper does not provide details on how the sentiment analysis was carried out. It doesn’t seem to provide convincing evidence of belief change.
- The best practices recommendation made in section 6.1 is rather conceptual. It can be very complicated to implement these ideas in general since there are many different use cases of the agents. It does not seem straightforward how to provide information on the limitations on agent capabilities. Even if such information can be available, it is also not obvious how the labelers can make sense of the information provided.

**Requested Changes:**

-	‘Using’ is used twice here: ‘We use these preferences to train the agent using using Contrastive Preference Learning’
-	The paper should provide justification and clarify the limitations on the assumptions made in the main theorem e.g., Theorem 4.3.
-	The paper should discuss whether the notion of the idea belief, which is defined implicitly through the best achievable post-RLHF policy under a fixed dataset,  can ever be approximated in practice.
-	The paper should provide evidence on whether it is possible to implement the potential best practices.
-	The paper should provide details on the sentiment analysis used in the human study.
-	The paper should justify more carefully why averaging repeated Likert responses is the right analysis choice, why only extremely confident responses are retained, and whether results are robust to alternative analyses.

---

> ### Author Response · Authors · 2026-04-04
> **Rebuttal to reviewer 5Bvg: RC1, RC2 and RC3**
>
> We thank Reviewer 5Bvg for the detailed and thoughtful review. We are glad the reviewer finds the belief-based framework for preference modeling in RLHF to be a novel contribution. We address the requested changes and weaknesses below.
>
> **RC1**: We will fix the typo it in the updated draft.
>
> **RC2**: We agree that a clearer discussion connecting Theorem 4.3 to broader settings is warranted.
>
> The tabular MDP and pairwise comparison setting used in Theorem 4.3 is a standard starting point for theoretical analysis in the RLHF literature (Zhu et al., 2023; Wang et al., 2023). Our theorem builds on this foundation to study the specific effect of belief disagreement on post-RLHF performance.
>
> Regarding the noiseless preference, deterministic policy, and preference-obeying assumptions: these correspond to the limit $\alpha \to \infty$ in the belief-based preference model. When $\alpha$ is finite (the realistic case), preferences are noisy. By the Luce choice axiom (Luce, 1959), the unique policy consistent with the resulting pairwise preferences is a softmax policy $\pi(a \mid s) \propto \exp(\alpha \, Q^{\pi_{\text{belief}}}(s, a))$, which is stochastic rather than deterministic. Under this relaxation, the Performance Difference Lemma (Kakade & Langford, 2002) still applies and yields a bound of the form:
>
> $$J^{\pi_\delta} \geq J^{\pi^\star_{\text{post}}} - \frac{2 \, d_\mu^{\pi_\delta}(s') \cdot A_{\max}(s') \cdot p(1-p)(e^{\alpha\delta} - 1)}{(1-\gamma)(1 + p(e^{\alpha\delta} - 1))}$$
>
> where $p = \pi^\star_{\text{post}}(a' \mid s')$, $d_\mu^{\pi_\delta}(s')$ is the discounted visitation probability of the disagreement state, and $A_{\max}(s') = \max_a |A^{\pi^\star_{\text{post}}}(s', a)|$ is the maximum advantage magnitude at $s'$ under the ideal post-RLHF policy. This recovers Theorem 4.3 as $\alpha \to \infty$. This generalized bound gives rough intuition for how performance could degrade under relaxed assumptions, though we acknowledge it is not tight.
>
> We emphasize that these quantities are used purely for theoretical bounding purposes — we do not suggest computing them in practice. Under our current assumptions, the bound simplifies to $\delta / (1-\gamma)$, which avoids dependence on $A_{\max}(s')$ and provides a clean characterization of how belief disagreement translates to performance loss.
>
> Regarding multiple disagreements: the bound naturally extends to $k$ disagreement states $s_1', s_2', \dots, s_k'$ via the visitation distribution $d_\mu^{\pi_\delta}(s_i')$, which captures how disagreements interact through the MDP. Disagreements at rarely-visited states contribute proportionally less to the performance loss. If a tighter bound on $d_\mu^{\pi_\delta}(s_i')$ is available (e.g., from knowledge of MDP structure), it can be substituted for a tighter result. In general, bounding each $d_\mu^{\pi_\delta}(s_i') \leq 1$ recovers a looser but assumption-free bound.
>
> The synthetic experiments in Section 5 were specifically designed to verify that the qualitative conclusions hold in more realistic cases where these assumptions are broken and where we cannot prove formal bounds — the theory and experiments are complementary in painting the full picture.
>
> We will include the generalized theorem, corollary, and full proofs in the appendix of the updated manuscript.
>
> *References:*
> - Luce, R.D. (1959). *Individual Choice Behavior: A Theoretical Analysis.* John Wiley & Sons.
> - Zhu, B., Jordan, M., & Jiao, J. (2023). Principled Reinforcement Learning with Human Feedback from Pairwise or K-wise Comparisons. *ICML 2023*.
> - Wang, Y., Liu, Q., & Jin, C. (2023). Is RLHF More Difficult than Standard RL? A Theoretical Perspective. *NeurIPS 2023*.
>
> **RC3**: We want to clarify that we never intend to compute or approximate $Q^{\pi^\star_{\text{belief}}}$ in practice. It is a theoretical quantity that serves as the reference point for defining agent-labeler disagreement and deriving the performance bound — deviations from these ideal beliefs are precisely what introduces error, which is what our theory bounds. Importantly, $Q^{\pi^\star_{\text{belief}}}$ is not derived from $\pi^\star_{\text{post}}$; rather, it represents accurate beliefs about the agent's capabilities, and providing labelers with such accurate beliefs is what recovers $\pi^\star_{\text{post}}$ — that is the intended direction of interpretation.
>
> This makes the problem practical: showing labelers on-policy demonstrations of the agent's current behavior gives them an informed understanding of agent capabilities, likely bringing their beliefs closer to $Q^{\pi^\star_{\text{belief}}}$. Generating such demonstrations is inexpensive, requiring only running the current policy, and well worth the cost compared to learning a sub-optimal policy from uninformed labels. We will clarify this interpretation in the updated manuscript.

---

> > ### Author Response · Authors · 2026-04-04
> > **Rebuttal to reviewer 5Bvg: RC4, RC5**
> >
> > **RC4: Evidence on implementability of best practices**: Our human study in Section 6 is itself evidence of implementability. The two recommendations in Section 6.1 are: (1) inform labelers about known limitations, and (2) online preference collection with intermittent priming. Both amount to the same core mechanism: show labelers what the agent can and cannot do before they provide preferences.
> >
> > Our human study implements exactly this — a single priming video of the agent's driving behavior was sufficient to produce statistically significant shifts in preference behavior (Dunn-Bonferroni p=0.015, Cliff's d=0.31). Each component of the proposed online loop — collect preferences, train the agent, show labelers rollouts from the updated policy, repeat — uses standard, existing tools. The only new step is showing demonstrations, which is inexpensive (just running the current policy) and well worth the cost compared to learning a sub-optimal policy from uninformed labels.
> >
> > We acknowledge that domain-specific challenges exist (e.g., what constitutes an informative demonstration for a language model agent), but the fundamental mechanism — that exposing labelers to agent behavior calibrates their beliefs and improves labeling — is empirically validated by our study. We will make this connection more explicit in the updated manuscript.
> >
> > **RC5: Details on sentiment analysis**: We apologize for this omission — details present in an earlier draft were inadvertently omitted during editing. We will restore them in the updated manuscript.
> >
> > After priming, participants in the safe and unsafe priming groups provided free-text feedback about the car's driving performance. To quantify these responses, we used GPT-4 (OpenAI et al., 2024) with the following prompt:
> >
> > > *"Analyze the following feedback about a car's driving skills and rate its safety from 1 to 10 stars. Are there any skills this car need to learn to be a more effective driver? Only consider the presence of any safety concerns and suggestions for improvement with small consideration for factors unrelated to safety: Feedback: [PARTICIPANT'S FEEDBACK] Provide a structured analysis with reasons for the assigned score."*
> >
> > GPT-4 was the strongest available model at the time our study was conducted. We chose GPT-4 over human sentiment annotators for greater consistency — LLM-based annotations have been shown to achieve higher internal consistency than human annotators for sentiment analysis tasks (Bojić et al., 2025; Zhang et al., 2024). Our task — rating safety sentiment of short free-text responses on a numerical scale — is straightforward relative to the benchmarks in that work.
> >
> > The sentiment analysis provides direct evidence that priming changed participants' beliefs about the car's capabilities (Figure 4): participants who saw the unsafe video described the car less favorably than those who saw the safe video. This establishes the first link in the causal chain — that priming shifts beliefs. The Kruskal-Wallis and Dunn-Bonferroni analysis on the Likert preference data then establishes the second link — that these shifted beliefs affect labeling behavior. Together, the two analyses confirm: priming → belief change (sentiment analysis) → preference change (Likert analysis).
> >
> > **References:**
> > - Zhang, W. et al. Sentiment Analysis in the Era of Large Language Models: A Reality Check. *Findings of the Association for Computational Linguistics: NAACL 2024*, 3881–3906.
> > - Bojić, L., et al. . Comparing Large Language Models and Human Annotators in Latent Content Analysis of Sentiment, Political Leaning, Emotional Intensity and Sarcasm. *Scientific Reports 2025*.

---

> > > ### Author Response · Authors · 2026-04-04
> > > **Rebuttal to reviewer 5Bvg: RC6 and additional weakness**
> > >
> > > **RC6: Statistical analysis choices:** We address each part:
> > >
> > > *Averaging repeated Likert responses.* Our design has three properties that together make standard tests difficult to apply directly: (1) more than two groups, (2) ordinal Likert data, and (3) repeated measures per participant. The reviewer suggests mixed-effects ordinal regression, specifically cumulative link mixed models (CLMM). CLMMs require the proportional odds assumption — that the effect of priming is uniform across all thresholds of the Likert scale. In our setting, priming may disproportionately affect certain parts of the scale (e.g., shifting responses at the extremes more than in the middle), and we did not have strong grounds to assume proportional odds holds. Additionally, with 46 participants per group and only high-confidence responses retained, the effective sample size may be insufficient for stable estimation of the random effects variance components in a CLMM.
> > >
> > > Averaging each participant's responses produces a single independent summary statistic per participant, enabling the Kruskal-Wallis test — a well-established nonparametric test that makes minimal distributional assumptions. While some within-participant information is lost, we opted for a test whose assumptions we could be confident about rather than a more powerful test whose assumptions may be violated.
> > >
> > > *Retaining only extremely confident responses.* Our experience with online data collection platforms such as Prolific is that responses given with low self-reported confidence frequently reflect participant disengagement or rushed responding rather than genuine uncertainty about the preference. This is compounded in our setting where participants must watch and compare two videos before answering — low confidence often signals insufficient attention to the stimuli. The confidence filter works alongside our attention-check filtering (removing participants who fail to prefer clearly-better trajectories) as a complementary quality control measure.
> > >
> > > *Robustness to alternative analyses.* We note that our current analysis is already conservative by design: the Bonferroni correction in the Dunn's test is the most stringent multiple comparisons correction available, and the Kruskal-Wallis test makes minimal distributional assumptions. Despite this conservatism, we still observe statistical significance at the 5% level (p=0.015 between safe and unsafe priming). We believe a less conservative analysis would only strengthen these results, not weaken them.
> > >
> > > **Weakness: synthetic experiments limited to epsilon-greedy noise**: We agree that we have not demonstrated the effect in more realistic settings — our synthetic experiments were designed as a controlled setup to isolate the effect of belief mismatch. Our paper builds the case incrementally: the human study in CARLA establishes that beliefs can be shifted and that this changes labeling behavior, the gridworld experiments show that belief mismatch degrades post-RLHF performance under controlled conditions, and the theory provides formal worst-case guarantees. Running the full RLHF pipeline with human labels would require a prohibitively large number of labeled preferences to train agents in realistic scenarios. Extending the empirical investigation to more realistic settings is a natural next step.

---

### Review · Reviewer_rrPM · 2026-03-31

**Summary Of Contributions:**

Summary:
This paper proposes a belief-based preference model for RLHF, arguing that human feedback is driven not only by objective rewards but also by internal assumptions about the agent’s future capabilities. By replacing the standard optimal advantage with a belief-contingent advantage within a regret-style framework, the authors formalize how a mismatch between human expectations and actual agent constraints defined as agent–labeler disagreement and how it leads to performance degradation. The work establishes a theoretical bound on this loss and provides empirical validation through both synthetic gridworld simulations and a human study in a driving simulator, demonstrating that priming a user’s perception of agent capability significantly and predictably shifts their preference behavior.


Strengths:
1. The paper identifies and formalizes a significant oversight in current RLHF modeling, the assumption that humans evaluate actions based on an optimal agent advantage. The proposed belief-based preference model provides a more realistic psychological foundation for how feedback is generated.
2. The hypothesis is well-supported across multiple axes, including a formal mathematical bound on agent-labeler disagreement, controlled synthetic experiments on capability constraints, and a human study demonstrating the effects of priming.
3. The authors provide concrete recommendations for practitioners to solve this issue

Weakness:
1. The formal bounds established in Theorem 4.3 rely on highly restrictive assumptions, specifically that the environment is tabular, the post-RLHF policy is deterministic, and the human preferences are entirely noiseless. While useful for intuition, these assumptions do not reflect RLHF settings involving function approximation and stochasticity.
2. The synthetic experiment used CPL with synthetic labels generated from Qπbelief. it does not test the entire RLHF pipeline with human labels or multiple domains to establish robustness.
3. In the human study, it is difficult to determine if priming altered specific beliefs about agent capability or if it simply shifted the participants' general risk tolerance and caution.

**Audience:**

Yes

**Audience Explanation:**

RLHF is currently critical and widely discussed alignment tools in the LLM era. Researchers working on reinforcement learning and the practical alignment of large language models will find this work useful. By addressing a fundamental flaw in standard preference model for RLHF this paper provides valuable insights.

**Claims And Evidence:**

Yes

**Claims Explanation:**

The authors provide strong evidence for their claims through mathematical theory, computational simulations, and empirical human studies.

**Requested Changes:**

Requested Changes:
1. Theorem 4.3 relies on strong assumptions. However, the synthetic GridWorld experiments utilize an epsilon-greedy policy and CPL. The authors must explicitly discuss how the bounds in Theorem 4.3 hold, relax, or fail to generalize when function approximation and stochastic policies are introduced.
2. Discuss about, How would you operationalize the normative ideal belief in practice? For example, in an online setting, could you estimate πpost* by periodically showing rollouts from the current policy and collecting belief-calibration judgments to fit πbelief?

---

> ### Author Response · Authors · 2026-04-04
> **Rebuttal to reviewer rrPM: requested changes and weaknesses**
>
> We thank Reviewer rrPM for the careful evaluation and for recognizing that the hypothesis is well-supported across theory, synthetic experiments, and the human study, as well as the concrete practitioner recommendations. We address the requested changes and weaknesses below.
>
> **RC1: Theorem 4.3 generalization to function approximation / stochastic policies:** Thank you for this important point. Our paper establishes the belief mismatch phenomenon at three levels: the CARLA human study (Section 6) validates that this effect exists with real humans, the synthetic gridworld experiments (Section 5) show that it impacts post-RLHF performance, and Theorem 4.3 provides a formal proof in a setting where a clean analytical bound is possible.
>
> The tabular MDP and pairwise comparison setting used in Theorem 4.3 is a standard starting point for theoretical analysis in the RLHF literature (Zhu et al., 2023; Wang et al., 2023). The noiseless preference, deterministic policy, and preference-obeying assumptions correspond to the limit $\alpha \to \infty$ in the belief-based preference model. When $\alpha$ is finite, the Luce choice axiom (Luce, 1959) implies a stochastic softmax policy, and we have derived a generalized version of Theorem 4.3 under this relaxation, which will be included in the updated manuscript. The generalized bound introduces dependence on $A_{\max}(s')$, the maximum advantage magnitude at the disagreement state. We emphasize that these quantities are used purely for theoretical bounding purposes — we do not suggest computing them in practice. Under our current assumptions, the bound simplifies to $\delta/(1-\gamma)$, providing a clean characterization of how belief disagreement translates to performance loss.
>
> Extending the bound to settings with function approximation, where a belief perturbation at one state can affect the policy everywhere through parameter sharing, remains an important direction for future work. We will add a discussion of these points to the updated manuscript.
>
> *References:*
> - Luce, R.D. (1959). *Individual Choice Behavior: A Theoretical Analysis.* John Wiley & Sons.
> - Zhu, B., Jordan, M., & Jiao, J. (2023). Principled Reinforcement Learning with Human Feedback from Pairwise or K-wise Comparisons. *ICML 2023*.
> - Wang, Y., Liu, Q., & Jin, C. (2023). Is RLHF More Difficult than Standard RL? A Theoretical Perspective. *NeurIPS 2023*.
>
> **RC2: Operationalizing the normative ideal belief in practice:** We want to first clarify that we do not suggest estimating $Q^{\pi^\star_{\text{belief}}}$ or $\pi^\star_{\text{post}}$ in practice — these quantities serve a theoretical role that yielded a principled argument for a simple practical intervention: showing labelers on-policy rollouts. $Q^{\pi^\star_{\text{belief}}}$ is not derived from $\pi^\star_{\text{post}}$; rather, it represents accurate beliefs about the agent's capabilities, and providing labelers with such beliefs is what recovers $\pi^\star_{\text{post}}$ — that is the intended direction of interpretation.
>
> We agree this can be done periodically in an online loop. Importantly, these demonstrations need not come from a post-RLHF policy — rollouts from the agent's current policy at any stage of training are sufficient to calibrate beliefs.
>
> **Weakness on not testing full RLHF pipeline with human labels**: We agree. Running the full RLHF pipeline with human labels would require a large number of labeled preferences to train agents in realistic scenarios, which is prohibitively costly for our current setup.
>
> **Weakness related to effect of priming on beliefs**: We appreciate this concern. We first apologize for omitting the sentiment analysis prompt while editing — these details will be restored in the updated manuscript.
>
> After priming, we asked participants specifically about the car's capabilities — which skills they think would be useful for the car to learn to become a more effective driver. This question is directed at their beliefs about the agent, not about risk in general. We then used GPT-4 to perform sentiment analysis on these responses with the following prompt:
>
> > *"Analyze the following feedback about a car's driving skills and rate its safety from 1 to 10 stars. Are there any skills this car need to learn to be a more effective driver? Only consider the presence of any safety concerns and suggestions for improvement with small consideration for factors unrelated to safety: Feedback: [PARTICIPANT'S FEEDBACK] Provide a structured analysis with reasons for the assigned score."*
>
> The resulting sentiment scores (Figure 4) show a clear separation between safe and unsafe priming groups on this car-specific question, suggesting that priming changed beliefs about the agent's capabilities specifically. Separately, the Likert preference data confirms these shifted beliefs affected labeling behavior (Dunn-Bonferroni p=0.015).

---

### Review · Reviewer_MXWP · 2026-04-01

**Summary Of Contributions:**

This paper studies a gap in RLHF: human labelers may judge a partial trajectory based not only on the observed segment, but also on what they believe the agent will be able to do after training. The paper replaces the standard optimal-advantage view with a belief-based preference model, defines a normative target tied to the best achievable post-RLHF policy over a fixed queried set, and proves that disagreement can reduce return. The paper then supports this framing with a GridWorld study, where performance is generally the best when labeler belief matches actual agent capability, and with a CARLA human study showing that simple priming changes the preferences people provide.

I think this is an interesting study that gives a fresh angle on RLHF and can attract interest from part of the community. A key strength is the clear connection between conceptual framing, theory, simulated experiments, and a human study. A main weakness is that the normative target may require oracle knowledge and is not directly computable in practice.

**Audience:**

Yes

**Audience Explanation:**

I expect this paper to interest readers working on RLHF, preference learning, reward learning, alignment, and human-AI interaction. The main reason is that it points to a simple but useful issue in preference collection: labelers may evaluate risky or efficient behaviors through an internal model of what the trained agent can later do, rather than through $A^*$ alone. When agent capability is capped by limited data, weak function approximation, regularization, or execution noise, the paper argues that the better normative target is the best achievable post-RLHF policy rather than absolute optimality. That is useful for both theory and data collection practice. I also think the paper can interest researchers outside core RLHF because it links preference data quality to human belief formation and survey design. The human study suggests that beliefs can be shifted in predictable ways through priming, which means that data collection procedures themselves may influence the reward signal used for learning.

**Broader Impact Concerns:**

I do not see an ethical concern.

**Claims And Evidence:**

Yes

**Claims Explanation:**

I think the core claims are supported for the scope the paper studies. On the theory side, the paper is clear about its assumptions and gives a clean formal result: if the labeler's belief differs from the ideal belief at a decision point, then the learned policy can lose return relative to the best achievable post-RLHF policy, and the size of that loss grows with the size of the disagreement and with how much future outcomes matter. This result does not solve the full practical problem, but it does clearly support the paper's main point that disagreement between labeler belief and agent capability can hurt RLHF.

The synthetic gridworld experiment is also consistent with this argument. The setup directly varies the mismatch between the labeler's assumed capability and the agent's actual capability, and the reported results show that returns are usually best when these two are aligned. The paper also gives a clear qualitative explanation for why optimistic labeler beliefs can be harmful: they encourage paths that are only safe for a highly capable agent, while a weaker agent may fail when trying to execute them. That supports the paper's main mechanism well.

**Requested Changes:**

- Please make the distinction between a normative target and a practical method much clearer. The paper's ideal labeler belief is defined as the belief that would lead to the best post-RLHF policy achievable under the fixed comparison set, but the paper also states that finding this ideal target may require oracle knowledge, such as knowledge of the environment or the learning rule.

- Add a short discussion of the difference between truthful calibration and manipulative priming. Informing labelers about real capability limits may improve preference quality, but the same tool could also be used to steer labels toward a researcher's preferred outcome.

---

> ### Author Response · Authors · 2026-04-04
> **Rebuttal to reviewer MXWP**
>
> We thank Reviewer MXWP for the positive assessment and for highlighting the clear connection between conceptual framing, theory, simulated experiments, and the human study as a key strength. We address the requested changes below.
>
> ---
>
> **RC1: Make the distinction between a normative target and a practical method much clearer:** Thank you for this suggestion. While $Q^{\pi^\star_{\text{belief}}}$ is presented in connection with $\pi^\star_{\text{post}}$, the ideal belief is not derived from $\pi^\star_{\text{post}}$. Rather, $Q^{\pi^\star_{\text{belief}}}$ represents accurate beliefs about the agent's capabilities. When labelers hold such accurate beliefs, they provide preferences that lead to $\pi^\star_{\text{post}}$ — that is the intended direction of interpretation. The normative ideal serves as a theoretical reference point for defining disagreement and deriving the performance bound. The practical method is separate and simple: show labelers on-policy demonstrations of the agent's current behavior to bring their beliefs closer to $Q^{\pi^\star_{\text{belief}}}$. We will make this distinction more explicit in the updated manuscript.
>
> **RC2: Discuss truthful calibration vs. manipulative priming:** Thank you for raising this concern. The key distinction is whether the demonstrations shown to labelers are representative of the agent's actual on-policy behavior. Truthful calibration shows unfiltered rollouts reflecting real capabilities and limitations; manipulative priming would involve curating demonstrations to create a misleading picture. We will add a discussion of this distinction and the associated ethical considerations to the updated manuscript.

---

### Decision · Action_Editor_TZkW · 2026-06-27

**Recommendation:** Accept as is

**Audience:**

Yes

**Audience Explanation:**

The paper will interest TMLR readers working on RLHF, preference learning, reward learning, alignment, and human-AI interaction. Its focus on how labelers’ beliefs about agent capability shape preferences is timely and useful.

**Claims And Evidence:**

Yes

**Claims Explanation:**

The main claims are sufficiently supported for acceptance. The paper makes a conceptual contribution by arguing that human preferences in RLHF depend not only on reward or optimal-value assumptions but also on labelers’ beliefs about the agent's capabilities. This claim is supported through three complementary components: a formal belief-based preference model and theoretical bounds, controlled synthetic experiments showing that belief-capability mismatch can degrade learned policy performance.

The evidence is not without limitations. Reviewer 5Bvg notes that the theorem relies on restrictive assumptions and that the empirical validation does not yet demonstrate gains in a full-scale practical RLHF pipeline. The rebuttal addresses the main concerns by clarifying the theorem’s role as a clean theoretical characterization, discussing stochastic/noisy extensions, framing the normative ideal as a theoretical reference rather than a directly computable target, and providing more detail on the human-study analysis. These limitations should be documented clearly in the final version.